# Deep Partial Updating

## Abstract

Emerging edge intelligence applications require the server to continuously retrain and update deep neural networks deployed on remote edge nodes to leverage newly collected data samples. Unfortunately, it may be impossible in practice to continuously send fully updated weights to these edge nodes due to the highly constrained communication resource. In this paper, we propose the weight-wise deep partial updating paradigm, which smartly selects only a subset of weights to update at each server-to-edge communication round, while achieving a similar performance compared to full updating. Our method is established through analytically upper-bounding the loss difference between partial updating and full updating, and only updates the weights which make the largest contributions to the upper bound. Extensive experimental results demonstrate the efficacy of our partial updating methodology which achieves a high inference accuracy while updating a rather small number of weights.

## 1 Introduction

To deploy deep neural networks (DNNs) on resource-constrained edge devices, extensive research has been done to compress a well-trained model via pruning (Han et al., 2016; Renda et al., 2020) and quantization (Courbariaux et al., 2015; Rastegari et al., 2016). During on-device inference, compressed networks may achieve a good balance between model performance (*e.g.*, prediction accuracy) and resource demand (*e.g.*, memory, computation, energy). However, due to the lack of relevant training data or an unknown sensing environment, pre-trained DNN models may not yield satisfactory performance. Retraining the model leveraging newly collected data (from edge devices or from other sources) is needed for desirable performance. Example application scenarios of relevance include vision robotic sensing in an unknown environment (*e.g.*, Mars) (Meng et al., 2017), local translators on mobile phones (Bhandare et al., 2019), and acoustic sensor networks deployed in Alpine environments (Meyer et al., 2019).

It is mostly impossible to perform on-device retraining on edge devices due to their resource-constrained nature. Instead, retraining often occurs on a remote server with sufficient resources. One possible strategy to continuously improve the model performance on edge devices is a two-stage iterative process: (*i*) at each round, edge devices collect new data samples and send them to the server, and (*ii*) the server retrains the network using all collected data, and then sends the updates to each edge device (Brown & Sreenan, 2006). An essential challenge herein is that the transmissions in the second stage are highly constrained by the limited communication resource (*e.g.*, bandwidth, energy) in comparison to the first stage. State-of-the-art DNN models always require tens or even hundreds of mega-Bytes (MB) to store parameters, whereas a single batch of data samples (a number of samples that can lead to reasonable updates in batch training) needs a relatively smaller amount of data. For example, for CIFAR10 dataset (Krizhevsky et al., 2009), the weights of a popular VGGNet require $56.09$MB storage, while one batch of 128 samples only uses around $0.40$MB (Simonyan & Zisserman, 2015; Rastegari et al., 2016). As an alternative, the server sends a full update once or rarely. But in this case, every node will suffer from a low performance until such an update occurs.

Besides, edge devices could decide on and send only critical samples by using active learning schemes (Ash et al., 2020). The server may also receive training data from other sources, *e.g.*, through data augmentation or new data collection campaigns. These considerations indicate that the updated weights which are sent to edge devices by the server at the second stage become a major bottleneck.

To resolve the above challenges pertaining to updating the network, we propose to partially update the network through changing only a small subset of the weights at each round. Doing so can significantly

reduce the server-to-device communication overhead. Furthermore, fewer parameter updates also lead to less memory access on edge devices, which in turn results in smaller energy consumption related to (compressed) full updating (Horowitz, 2014). Our goal of performing partial updating is to determine which subset of weights shall be updated at each round, such that a similar accuracy can be achieved compared to fully updating all weights.

Our key concept for partial updating is based on the hypothesis, that *a weight shall be updated only if it has a large contribution to the loss reduction* given the newly collected data samples. Specially, we define a binary mask $m$ to describe which weights are subject to update, *i.e.*, $m_i = 1$ implies updating this weight and $m_i = 0$ implies fixing the weight to its initial value. For any $m$, we establish an analytical upper bound on the difference between the loss value under partial updating and that under full updating. We determine an optimized mask $m$ by combining two different view points: (*i*) measuring the "global contribution" of each weight to the upper bound through computing the Euclidean distance, and (*ii*) measuring each weight's "local contribution" within each optimization step using gradient-related information. The weights to be updated according to $m$ will be further sparsely fine-tuned while the remaining weights are rewound to their initial values.

**Related Work.** Although partial updating has been adopted in some prior works, it is conducted in a fairly coarse-grained manner, *e.g.*, layer-wise or neuron-wise, and targets at completely different objectives. Especially, under continual learning settings, (Yoon et al., 2018; Jung et al., 2020) propose to freeze all weights related to the neurons which are more critical in performing prior tasks than new ones, to preserve existing knowledge. Under adversarial attack settings, (Shokri & Shmatikov, 2015) updates the weights in the first several layers only, which yield a dominating impact on the extracted features, for better attack efficacy. Under architecture generalization settings, (Chatterji et al., 2020) studies the generalization performance through the resulting loss degradation when rewinding the weights of each individual layer to their initial values. Unfortunately, such techniques cannot be applied in our problem setting which seeks a fine-grained, *i.e.*, weight-wise, partial updating given newly collected training samples in an iterative manner.

The communication cost could also be reduced through some other techniques, *e.g.*, quantizing/encoding the updated weights and the transmission signal. But note that these techniques are orthogonal to our approach and could be applied in addition. Also note that our defined partial updating setting differs from the communication-efficient distributed (federated) training settings (Lin et al., 2018; Kairouz et al., 2019), which study how to compress multiple gradients calculated on different sets of non-*i.i.d.* local data, such that the aggregation of these (compressed) gradients could result in a similar convergence performance as centralized training on all data.

Traditional pruning methods (Han et al., 2016; Frankle & Carbin, 2019; Renda et al., 2020) aim at reducing the number of operations and storage consumption by setting some weights to zero. Sending a pruned network (non-zero's weights) may also reduce the communication cost, but to a much lesser extent as shown in the experimental results, see Section 4.4. In addition, since our objective namely reducing the server-to-edge communication cost when updating the deployed networks is fundamentally different from pruning, we can leverage some learned knowledge by retaining previous weights (*i.e.*, partial updating) instead of zero-outing (*i.e.*, pruning).

**Contributions.** Our contributions can be summarized as follows.

- We formalize the deep partial updating paradigm, *i.e.*, how to iteratively perform weight-wise partial updating of deep neural networks on remote edge devices if newly collected training samples are available at the server. This substantially reduces the computation and communication demand on the edge devices.
- We propose a new approach that determines the optimized subset of weights that shall be selected for partial updating, through measuring each weight's contribution to the analytical upper bound on the loss reduction.
- Experimental results on three popular vision datasets show that under the similar accuracy level, our approach can reduce the size of the transmitted data by $91.7\%$ on average (up to $99.3\%$), namely can update the model averagely 12 times more frequent than full updating.

## 2 NOTATION AND SETTING

In this section, we define the notation used throughout this paper, and provide a formalized problem setting, *i.e.*, deep partial updating. We consider a set of remote edge devices that implement on-device

inference. They are connected to a host server that is able to perform network training and retraining. We consider the necessary amount of information that needs to be communicated to each edge device to update its inference network.

Assume there are in total $R$ rounds of network updates. The network deployed in the $r^{\text{th}}$ round is represented with its weight vector $\boldsymbol{w}^r$. The training data used to update the network for the $r^{\text{th}}$ round is represented as $\mathcal{D}^r = \delta\mathcal{D}^r \cup \mathcal{D}^{r-1}$. In other words, newly collected data samples $\delta\mathcal{D}^r$ are made available to the server in round $r - 1$.

To reduce the amount of information that needs to be sent to edge devices, only partial weights of $\boldsymbol{w}^{r-1}$ shall be updated when determining $\boldsymbol{w}^r$. The overall optimization problem for weight-wise partial updating in round $r - 1$ can thus be formulated as

$$\min_{\delta\boldsymbol{w}^r} \quad \ell\left(\boldsymbol{w}^{r-1} + \delta\boldsymbol{w}^r; \mathcal{D}^r\right) \tag{1}$$

$$\text{s.t.} \quad \|\delta\boldsymbol{w}^r\|_0 \leq k \cdot I \tag{2}$$

where $\ell$ denotes the loss function, $\|.\|_0$ denotes the L0-norm, $k$ denotes the defined updating ratio which is closely related to the communication demand between server and edge devices, and $\delta\boldsymbol{w}^r$ denotes the increment of $\boldsymbol{w}^{r-1}$. Note that both $\boldsymbol{w}^{r-1}$ and $\delta\boldsymbol{w}^r$ are drawn from $\mathbb{R}^I$, where $I$ denotes the total number of weights.

In this case, only a fraction of $k \cdot I$ weights and the corresponding index information need to be communicated to each edge device for updating the network in round $r$, namely the partial updates $\delta\boldsymbol{w}^r$. It is worth noting that the index information is relatively small in size compared to the partially updated weights (see Section 4). On each edge device, the weight vector is updated as $\boldsymbol{w}^r = \boldsymbol{w}^{r-1} + \delta\boldsymbol{w}^r$. To simplify the notation, we will only consider a single update, *i.e.*, from weight vector $\boldsymbol{w}$ (corresponding to $\boldsymbol{w}^{r-1}$) to weight vector $\widetilde{\boldsymbol{w}}$ (corresponding to $\boldsymbol{w}^r$) with

$$\widetilde{\boldsymbol{w}} = \boldsymbol{w} + \widetilde{\delta\boldsymbol{w}}$$

## 3 Partial Updating

We develop a two-step approach for resolving the partial updating optimization problem in Eq.(1)-Eq.(2). The final implementation used for the experimental results, see Section 4, contains some minor adaptations that do not change the main principles as explained next. In the first step, we compute a subset of all weights with only $k \cdot I$ weights. These weights will be allowed to change their values. In the second step, we optimize the weights in the chosen subset (considering the constraint of Eq.(2)) to minimize the loss function in Eq.(1). The overall approach is depicted in Figure 1.

The approach for the first step not only determines the subset of weights but also computes the initial values for the second (sparse) optimization step. In particular, we first optimize the loss function Eq.(1) from initial weights $\boldsymbol{w}$ with a standard optimizer, *e.g.*, SGD or its variants. As a result, we obtain the minimized loss $\ell\left(\boldsymbol{w}^{\text{f}}\right)$ with $\boldsymbol{w}^{\text{f}} = \boldsymbol{w} + \delta\boldsymbol{w}^{\text{f}}$, where the superscript f denotes "full updating". To consider the constraint Eq.(2), the information gathered during this optimization is used to determine the subset of weights that will be changed and therefore, that need to be communicated to the edge devices.

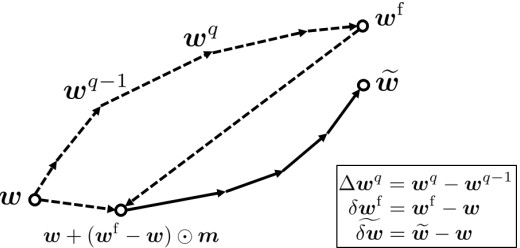

$$\Delta\boldsymbol{w}^q = \boldsymbol{w}^q - \boldsymbol{w}^{q-1}$$
$$\delta\boldsymbol{w}^{\text{f}} = \boldsymbol{w}^{\text{f}} - \boldsymbol{w}$$
$$\widetilde{\delta\boldsymbol{w}} = \widetilde{\boldsymbol{w}} - \boldsymbol{w}$$

Figure 1: The figure depicts the overall approach that consists of two steps. The first step is depicted with dotted arrows and starts from the deployed network weights $\boldsymbol{w}$. In $Q$ steps, the network is optimized which results in weights $\boldsymbol{w}^{\text{f}}$. Based on the collected information, a mask $\boldsymbol{m}$ is determined that characterizes the set of weights that are rewound to the ones of $\boldsymbol{w}$. Therefore, the initial solution for the second step has weights $\boldsymbol{w} + \delta\boldsymbol{w}^{\text{f}} \odot \boldsymbol{m}$. This initial solution is further optimized to the new weights $\widetilde{\boldsymbol{w}}$ by only changing weights that are allowed according to the mask, *i.e.*, $\widetilde{\delta\boldsymbol{w}}$ has only nonzero elements where the mask is 1.

In the explanation of the method in Section 3.1, we use the mask $\boldsymbol{m}$ with $\boldsymbol{m} \in \{0, 1\}^I$ to describe which weights are subject to change and which ones are not. The weights with $m_i = 1$ are trainable, whereas the weights with $m_i = 0$ will be rewound from the values in $\boldsymbol{w}^{\mathrm{f}}$ to their initial values in $\boldsymbol{w}$, *i.e.*, unchanged. Obviously, we find $\|\boldsymbol{m}\|_0 = \sum_i m_i = k \cdot I$. In summary, the purpose of this first step is to determine an optimized mask $\boldsymbol{m}$.

In the second step we start a weight optimization from a network with $k \cdot I$ weights from the optimized network $\boldsymbol{w}^{\mathrm{f}}$ and $(1 - k) \cdot I$ weights from the previous, still deployed network $\boldsymbol{w}$. In other words, the initial weights for this optimization are $\boldsymbol{w} + \delta\boldsymbol{w}^{\mathrm{f}} \odot \boldsymbol{m}$, where $\odot$ denotes an element-wise multiplication. We still use a standard optimizer. To determine the final solution $\widetilde{\boldsymbol{w}} = \boldsymbol{w} + \widetilde{\delta\boldsymbol{w}}$, we conduct a sparse fine-tuning, *i.e.*, we keep all weights with $m_i = 0$ constant during the optimization. Therefore, $\widetilde{\delta\boldsymbol{w}}$ is zero wherever $m_i = 0$, and only weights where $m_i = 1$ are updated.

## 3.1 METRICS FOR REWINDING

We will now describe a new metric that allows us to determine the weights that should be kept constant, *i.e.*, those whose masks satisfy $m_i = 0$. Like most learning methods, we focus on minimizing a loss function, since the loss is a more general metric than, for example, accuracy and perplexity, which are the metrics only used for classification and language modeling respectively. But we still report the other metrics in the evaluation. The two-step approach relies on the following assumption: the better the loss $\ell(\boldsymbol{w} + \delta\boldsymbol{w}^{\mathrm{f}} \odot \boldsymbol{m})$ of the initial solution for the second step, the better the final loss $\ell(\widetilde{\boldsymbol{w}})$. Therefore, the first step in the method should select a mask $\boldsymbol{m}$ such that the loss difference $\ell(\boldsymbol{w} + \delta\boldsymbol{w}^{\mathrm{f}} \odot \boldsymbol{m}) - \ell(\boldsymbol{w}^{\mathrm{f}})$ is as small as possible.

We will determine an optimized mask $\boldsymbol{m}$ by combining two different view points. The "global contribution" uses information contained in the difference $\delta\boldsymbol{w}^{\mathrm{f}}$ between the initial weights $\boldsymbol{w}$ and the optimized weights $\boldsymbol{w}^{\mathrm{f}}$ by the first step, namely the norm of incremental weights. The "local contribution" takes into account some gradient-based information that is gathered during the optimization in the first step, *i.e.*, in the path from $\boldsymbol{w}$ to $\boldsymbol{w}^{\mathrm{f}}$. Both kinds of information will be combined to determine an optimized mask $\boldsymbol{m}$.

The two view points are based on the concept of smooth differentiable functions, see for example (Nesterov, 1998). A function $f(x)$ with $f : \mathbb{R}^d \rightarrow \mathbb{R}$ is called $L$-smooth if it has a Lipschitz continuous gradient $g(x)$: $\|g(x) - g(y)\|_2 \leq L\|x - y\|_2$ for all $x, y$. Note that Lipschitz continuity of a gradient is essential to ensuring convergence of many gradient-based algorithms. Under such a condition, one can derive the following bounds, see also (Nesterov, 1998):

$$|f(y) - f(x) - g(x)^{\mathrm{T}} \cdot (y - x)| \leq L/2 \cdot \|y - x\|_2^2 \quad \forall x, y \tag{3}$$

This basic relation is used to justify the global and the local contributions, *i.e.*, the rewinding metrics.

**Global Contribution.** Following some state-of-the-art methods for pruning, one would argue that a large absolute value in $\delta\boldsymbol{w}^{\mathrm{f}} = \boldsymbol{w}^{\mathrm{f}} - \boldsymbol{w}$ indicates that this weight has moved far from its initial value in $\boldsymbol{w}$. This motivates us to adopt the widely used unstructured magnitude pruning to solve the problem of determining an optimized mask $\boldsymbol{m}$. Magnitude pruning prunes the weights with the lowest magnitudes in a network, which is the current best-performed pruning method aiming at the trade-off between the model accuracy and the number of zero's weights (Renda et al., 2020).

Using $a - b \leq |a - b|$, Eq.(3) can be reformulated as $f(y) - f(x) - g(x)^T(y - x) \leq |f(y) - f(x) - g(x)^T(y - x)| \leq L/2 \cdot \|y - x\|_2^2$. Thus, we can bound the relevant difference in the loss $\ell(\boldsymbol{w} + \delta\boldsymbol{w}^{\mathrm{f}} \odot \boldsymbol{m}) - \ell(\boldsymbol{w}^{\mathrm{f}}) \geq 0$ as

$$\ell(\boldsymbol{w} + \delta\boldsymbol{w}^{\mathrm{f}} \odot \boldsymbol{m}) - \ell(\boldsymbol{w}^{\mathrm{f}}) \leq \boldsymbol{g}(\boldsymbol{w}^{\mathrm{f}})^{\mathrm{T}} \cdot \left(\delta\boldsymbol{w}^{\mathrm{f}} \odot (\boldsymbol{m} - \boldsymbol{1})\right) + L/2 \cdot \|\delta\boldsymbol{w}^{\mathrm{f}} \odot (\boldsymbol{m} - \boldsymbol{1})\|_2^2 \tag{4}$$

where $\boldsymbol{g}(\boldsymbol{w}^{\mathrm{f}})$ denotes the gradient of the loss function at $\boldsymbol{w}^{\mathrm{f}}$, and $\boldsymbol{1}$ is a vector whose elements are all 1. As the loss is optimized at $\boldsymbol{w}^{\mathrm{f}}$, *i.e.*, $\boldsymbol{g}(\boldsymbol{w}^{\mathrm{f}}) \approx \boldsymbol{0}$, we can assume that the gradient term is much smaller than the norm of the weight differences in Eq.(4). Therefore, we obtain approximately

$$\ell(\boldsymbol{w} + \delta\boldsymbol{w}^{\mathrm{f}} \odot \boldsymbol{m}) - \ell(\boldsymbol{w}^{\mathrm{f}}) \lesssim L/2 \cdot \|\delta\boldsymbol{w}^{\mathrm{f}} \odot (\boldsymbol{1} - \boldsymbol{m})\|_2^2 \tag{5}$$

The right hand side is clearly minimized if $m_i = 1$ for the largest absolute values of $\delta\boldsymbol{w}^{\mathrm{f}}$. This information is captured in the contribution vector

$$\boldsymbol{c}^{\mathrm{global}} = \delta\boldsymbol{w}^{\mathrm{f}} \odot \delta\boldsymbol{w}^{\mathrm{f}} \tag{6}$$

as $\mathbf{1}^{\mathrm{T}} \cdot \left(\boldsymbol{c}^{\mathrm{global}} \odot (\mathbf{1} - \boldsymbol{m})\right) = \|\delta \boldsymbol{w}^{\mathrm{f}} \odot (\mathbf{1} - \boldsymbol{m})\|_2^2$.

In summary, the $k \cdot I$ weights with the largest values in $\boldsymbol{c}^{\mathrm{global}}$ are assigned to mask values $m_i = 1$ and are further fine-tuned in the second step, whereas all others are rewound from $\boldsymbol{w}^{\mathrm{f}}$, and keep their initial values in $\boldsymbol{w}$. The pseudocode of Alg. 1 in Appendix A.1 shows this first approach.

**Local Contribution.** As experiments show, one can do better when using in addition some gradient-based information gathered during the first step, *i.e.*, optimizing the initial weights $\boldsymbol{w}$ in $Q$ traditional optimization steps, $\boldsymbol{w} = \boldsymbol{w}^0 \rightarrow \cdots \rightarrow \boldsymbol{w}^{q-1} \rightarrow \boldsymbol{w}^q \rightarrow \cdots \rightarrow \boldsymbol{w}^Q = \boldsymbol{w}^{\mathrm{f}}$.

Using $-a + b \leq |a - b|$, Eq.(3) can be reformulated as $f(x) - f(y) + g(x)^T(y - x) \leq |f(y) - f(x) - g(x)^T(y - x)| \leq L/2 \cdot \|y - x\|_2^2$. This leads us to bound each optimization step as

$$\ell(\boldsymbol{w}^{q-1}) - \ell(\boldsymbol{w}^q) \leq -\boldsymbol{g}(\boldsymbol{w}^{q-1})^{\mathrm{T}} \cdot \Delta \boldsymbol{w}^q + L/2 \cdot \|\Delta \boldsymbol{w}^q\|_2^2 \tag{7}$$

where $\Delta \boldsymbol{w}^q = \boldsymbol{w}^q - \boldsymbol{w}^{q-1}$. For a conventional gradient descent optimizer with a small learning rate we can use the approximation $|\boldsymbol{g}(\boldsymbol{w}^{q-1})^{\mathrm{T}} \cdot \Delta \boldsymbol{w}^q| \gg \|\Delta \boldsymbol{w}^q\|_2^2$ and obtain $\ell(\boldsymbol{w}^{q-1}) - \ell(\boldsymbol{w}^q) \lesssim -\boldsymbol{g}(\boldsymbol{w}^{q-1})^{\mathrm{T}} \cdot \Delta \boldsymbol{w}^q$. Summing up over all optimization iterations yields approximately

$$\ell(\boldsymbol{w}^{\mathrm{f}} - \delta \boldsymbol{w}^{\mathrm{f}}) - \ell(\boldsymbol{w}^{\mathrm{f}}) \lesssim -\sum_{q=1}^{Q} \boldsymbol{g}(\boldsymbol{w}^{q-1})^{\mathrm{T}} \cdot \Delta \boldsymbol{w}^q \tag{8}$$

Note that we have $\boldsymbol{w} = \boldsymbol{w}^{\mathrm{f}} - \delta \boldsymbol{w}^{\mathrm{f}}$ and $\delta \boldsymbol{w}^{\mathrm{f}} = \sum_{q=1}^{Q} \Delta \boldsymbol{w}^q$. Therefore, with $\boldsymbol{m} \sim \mathbf{0}$ we can reformulate Eq.(8) as $\ell\left(\boldsymbol{w} + \delta \boldsymbol{w}^{\mathrm{f}} \odot \boldsymbol{m}\right) - \ell(\boldsymbol{w}^{\mathrm{f}}) \lesssim \mathrm{U}(\boldsymbol{m})$ with the upper bound $\mathrm{U}(\boldsymbol{m}) = -\sum_{q=1}^{Q} \boldsymbol{g}(\boldsymbol{w}^{q-1})^{\mathrm{T}} \cdot \left(\Delta \boldsymbol{w}^q \odot (\mathbf{1} - \boldsymbol{m})\right)$ where we suppose that the gradients are approximately constant for small $\boldsymbol{m}$. Therefore, an approximate incremental contribution of each weight dimension to the upper bound on the loss difference $\ell\left(\boldsymbol{w} + \delta \boldsymbol{w}^{\mathrm{f}} \odot \boldsymbol{m}\right) - \ell(\boldsymbol{w}^{\mathrm{f}})$ can be determined by the negative gradient vector at $\boldsymbol{m} = \mathbf{0}$, denoted as

$$\boldsymbol{c}^{\mathrm{local}} = -\frac{\partial \mathrm{U}(\boldsymbol{m})}{\partial \boldsymbol{m}} = -\sum_{q=1}^{Q} \boldsymbol{g}(\boldsymbol{w}^{q-1}) \odot \Delta \boldsymbol{w}^q \tag{9}$$

This term is used to model the accumulated contribution of each weight to the overall loss reduction.

**Combining Global and Local Contribution.** So far, we independently calculate the global and local contributions $c^{\mathrm{global}}$ and $c^{\mathrm{local}}$, respectively. To avoid the impact due to the scale, we first normalize each contribution by its significance in its own set (either global contribution set or local contribution set). We conduct experiments on how to combine both normalized contributions, *e.g.*, taking the minimum. Interestingly, the most straightforward combination (*i.e.*, the sum of both normalized metrics) yields a better and more stable performance. Intuitively, local contribution can better identify critical weights w.r.t. the loss during training, while global contribution may be more robust for a highly non-convex loss landscape. Both metrics may be necessary when selecting weights to rewind. Therefore, the total contribution is computed as

$$\boldsymbol{c} = \frac{1}{\mathbf{1}^{\mathrm{T}} \cdot \boldsymbol{c}^{\mathrm{global}}} \boldsymbol{c}^{\mathrm{global}} + \frac{1}{\mathbf{1}^{\mathrm{T}} \cdot \boldsymbol{c}^{\mathrm{local}}} \boldsymbol{c}^{\mathrm{local}} \tag{10}$$

and $m_i = 1$ for the $k \cdot I$ largest values of $\boldsymbol{c}$ and $m_i = 0$ otherwise. The pseudocode of the corresponding algorithm is shown in Alg. 2 in Appendix A.2.

## 3.2 (RE-)INITIALIZATION OF WEIGHTS

In this section, we discuss the initialization of our method. $\mathcal{D}^1$ denotes the initial dataset used to train the network $\boldsymbol{w}^1$ from a randomly initialized network $\boldsymbol{w}^0$. $\mathcal{D}^1$ corresponds to the available dataset before deployment, or collected in the $0^{\mathrm{th}}$ round if there are no data available before deployment. $\{\delta \mathcal{D}^r\}_{r=2}^{R}$ denotes newly collected samples in each subsequent round.

Experimental results show (see Appendix D.1) that starting from a randomly initialized network can yield a higher accuracy after several rounds, compared to always training from the last round with weights $\boldsymbol{w}^{r-1}$. As a possible explanation, the optimizer could end in a hard to escape region of the search space if always trained from the last round for a long sequence of rounds. Thus, we propose to

re-initialize the weights after a certain number of rounds. In such a case, Alg. 2 does not start from the previous weights $\boldsymbol{w}^{r-1}$ but from randomly initialized weights. The re-initialized random network can be efficiently sent to the edge devices via a single random seed. The device can determine the weights by means of a random generator. This is a communication-efficient way of a random shift from a hard to escape region of the search space in comparison to other alternatives, such as learning to increase the loss or using the (averaged) weights in the previous rounds, as these fully changed weights need to be sent to each node. Each time the network is randomly re-initialized, the new partially updated network may suffer from an accuracy drop. However, we can simply avoid such an accuracy drop by not updating the network if the validation accuracy does not increase compared to the last round, see more details in Appendix D.2. Note that the learned knowledge thrown away by the re-initialization can be re-learned afterwards, since any newly collected samples are continuously stored and accumulated in the server. This also makes our setting different from continual learning, which aims at avoiding catastrophic forgetting without accessing (at least not all) old data.

To determine after how many rounds the network needs to be re-initialized, we conduct extensive experiments on different partial updating settings, see more discussions and results in Appendix D.2. In conclusion, the network is randomly re-initialized as long as the number of total newly collected data samples exceeds the number of samples when the network was re-initialized last time. For example, assume that at round $r$ the model is randomly (re-)initialized and partially updated from this random network on dataset $\mathcal{D}^r$. Then, the model will be re-initialized at round $r + n$, if $|\mathcal{D}^{r+n}| > 2 \cdot |\mathcal{D}^r|$. In the following, we use Deep Partial Updating (DPU) to present rewinding according to the combined contribution to the loss reduction (*i.e.*, Alg. 2) with the above re-initialization scheme.

## 4 EVALUATION

We implement DPU with Pytorch (Paszke et al., 2017), and evaluate its performance on multiple vision datasets, including MNIST (LeCun & Cortes, 2010), CIFAR10 (Krizhevsky et al., 2009), ILSVRC12 (ImageNet) (Russakovsky et al., 2015) using multilayer perceptron (MLP), VGGNet (Courbariaux et al., 2015; Rastegari et al., 2016), ResNet34 (He et al., 2016), respectively. We randomly select 30% of each original test dataset (original validation dataset for ILSVRC12) as the validation dataset, and the remainder as the test dataset. Let $|.|$ denote the number of samples in the dataset. Let $\{|\mathcal{D}^1|, |\delta\mathcal{D}^r|\}$ represent the available data samples along rounds, where $|\delta\mathcal{D}^r|$ is supposed to be constant along rounds. Both $\mathcal{D}^1$ and $\delta\mathcal{D}^r$ are randomly drawn from the original training dataset (only for evaluation purposes). For all pre-processing and random initialization, we apply the tools provided in Pytorch. We use the average cross-entropy as the loss function without a regularization term for better studying the effect on the training error caused by newly added data samples. We use Adam variant of SGD as the optimizer, except that Nesterov SGD is used for ResNet34 following the suggestions in (Renda et al., 2020). The test accuracy is reported, when the validation dataset achieves the highest Top-1 accuracy. When the validation accuracy does not increase compared to the last round, the model will not be updated to reduce communication overhead. This strategy is applied in all methods for a fair comparison. More implementation details are provided in Appendix C. We will open-source the code upon acceptance.

**One-shot Rewinding vs Iterative Rewinding.** Based on previous experiments on pruning (Renda et al., 2020), iterative pruning with retraining may yield a higher accuracy compared to one-shot pruning, yet requiring several times more optimization iterations and also extra handcrafted hyperparameters (*e.g.*, pruning ratio schedule). This paper focuses on comparing the performance of DPU with other baselines including full updating given the same number of optimization iterations per round. Thus, we conduct one-shot rewinding at each round, *i.e.*, the rewinding is executed only once to achieve the desired sparsity (as shown in Alg. 2).

**Indexing.** DPU generates a sparse tensor. In addition to the updated weights, the indices of these weights also need to be sent to each edge device. A simple implementation is to send the mask $\boldsymbol{m}$. $\boldsymbol{m}$ is a binary vector with $I$ elements, which are assigned with 1 if the corresponding weights are updated. Let $S_w$ denote the bitwidth of each single weight, and $S_x$ denote the bitwidth of each index. Directly sending $\boldsymbol{m}$ yields an overall communication cost of $I \cdot k \cdot S_w + I \cdot S_x$ with $S_x = 1$.

To save the communication cost on indexing, we further encode $\boldsymbol{m}$. Suppose that $\boldsymbol{m}$ is a random binary vector with a probability of $k$ to contain 1. The optimal encoding scheme according to Shannon yields $S_x(k) = k \cdot \log(1/k) + (1 - k) \cdot \log(1/(1 - k))$. Coding schemes such as Huffman

block coding can come close to this bound. Partial updating results in a smaller communication data size than full updating, if $S_w \cdot I > S_w \cdot k \cdot I + S_x(k) \cdot I$. Under the worst case for indexing cost, *i.e.*, $S_x(k = 0.5) = 1$, as long as $k < (32 - 1)/32 = 0.97$, partial updating can yield a smaller communication data size with $S_w = 32$-bit weights. In the following experiments, we will use $S_w \cdot k \cdot I + S_x(k) \cdot I$ to report the size of data transmitted from server to each node at each round, contributed by the partially updated weights plus the encoded indices of these weights.

## 4.1 Ablation Study of Rewinding Metrics

**Settings.** We first conduct a set of ablation experiments regarding different metrics of rewinding discussed in Section 3.1. We compare the influence of the local and global contributions as well as their combination, in terms of the incremental training loss caused by rewinding. The original VGGNet and ResNet34 are fully trained on a randomly selected dataset of $10^3$ and $4 \times 10^5$ samples, respectively. We execute full updating, *i.e.*, the first step of our approach, after adding $10^3$ and $2 \times 10^5$ new randomly selected samples, respectively. Afterwards, we conduct one-shot rewinding with all three metrics, *i.e.*, global contribution, local contribution, and combined contribution. Each experiment is conducted for a single round. We report the results over five runs.

**Results.** The training loss (mean $\pm$ standard deviation) after full updating (*i.e.*, $\ell(\boldsymbol{w}^\mathrm{f})$) and after rewinding (*i.e.*, $\ell(\boldsymbol{w} + \delta\boldsymbol{w}^\mathrm{f} \odot \boldsymbol{m})$) with three metrics is reported in Table 1. Note that these loss values are only intermediate results during partial updating. As seen in the table, the combined contribution always yields a lower or similar training loss after rewinding compared to the other two metrics. The smaller deviation also indicates that adopting the combined contribution yields more robust results. This validates the effectiveness of our proposed metric, *i.e.*, the combined contribution to the analytical upper bound on loss reduction.

Table 1: Comparing the training loss after rewinding according to different metrics.

| Benchmark | $k$ | Training loss | | | |
|---|---|---|---|---|---|
| | | Full updating | Global | Local | Combined |
| VGGNet (CIFAR10) | 0.01 | $0.086 \pm 0.001$ | $3.042 \pm 0.068$ | $\textbf{2.588} \pm \textbf{0.084}$ | $2.658 \pm 0.086$ |
| | 0.05 | | $2.509 \pm 0.056$ | $1.799 \pm 0.104$ | $\textbf{1.671} \pm \textbf{0.062}$ |
| | 0.1 | | $2.031 \pm 0.046$ | $1.337 \pm 0.076$ | $\textbf{0.994} \pm \textbf{0.034}$ |
| | 0.2 | | $1.196 \pm 0.049$ | $0.739 \pm 0.031$ | $\textbf{0.417} \pm \textbf{0.009}$ |
| ResNet34 (ILSVRC12) | 0.01 | $1.016 \pm 0.000$ | $3.340 \pm 0.109$ | $4.222 \pm 0.156$ | $\textbf{3.179} \pm \textbf{0.052}$ |
| | 0.05 | | $2.005 \pm 0.064$ | $2.346 \pm 0.036$ | $\textbf{1.844} \pm \textbf{0.022}$ |
| | 0.1 | | $1.632 \pm 0.044$ | $2.662 \pm 0.048$ | $\textbf{1.609} \pm \textbf{0.025}$ |
| | 0.2 | | $1.331 \pm 0.016$ | $3.626 \pm 0.062$ | $\textbf{1.327} \pm \textbf{0.008}$ |

## 4.2 Evaluation on Different Benchmarks

**Settings.** To the best of our knowledge, this is the first work on studying weight-wise partial updating a network using newly collected data in iterative rounds. Therefore, we developed three baselines for comparison, including (*i*) full updating (FU), where at each round the network is fully updated with a random initialization (*i.e.*, training from scratch, which yields a better performance as discussed in Section 3.2); (*ii*) random partial updating (RPU), where the network is trained from $\boldsymbol{w}^{r-1}$, while we randomly fix each layer's weights with a ratio of $(1 - k)$ and sparsely fine-tune the rest; and (*iii*) global contribution partial updating (GCPU), where the network is trained with Alg. 1 without re-initialization described in Section 3.2. Note that (*iii*) is extended from a state-of-the-art unstructured pruning method (Renda et al., 2020). The experiments are conducted with different types of networks on different benchmarks as mentioned earlier.

**Results.** We report the test accuracy of the network $\boldsymbol{w}^r$ along rounds in Figure 2. All methods start from the same $\boldsymbol{w}^0$, an entirely randomly initialized network. As seen in this figure, DPU clearly yields the highest accuracy in comparison to the other partial updating schemes on different benchmarks. For example, DPU can yield a final Top-1 accuracy of 92.85% on VGGNet, even exceeds the accuracy (92.73%) of full updating, while GCPU and RPU only acquire 91.11% and 82.21% respectively. In addition, we compare three partial updating schemes in terms of the accuracy difference related to full updating averaged over all rounds, and the ratio of the communication cost over all rounds related

to full updating in Table 2. As seen in the table, DPU reaches a similar or even higher accuracy as full updating, while incurring significantly fewer transmitted data sent from the server to each edge node. Specially, DPU saves around $99.3\%$, $98.2\%$ and $77.7\%$ of transmitted data on MLP, VGGNet, and ResNet34, respectively ($91.7\%$ in average). The communication cost ratios shown in Table 2 differ a little even for the same updating ratio $k$. This is because if the validation accuracy does not increase compared to the last round, the model will not be updated to reduce the communication overhead (also see the first paragraph of Section 4). We also report the number of updated rounds in Table 2.

We further investigate the benefit due to DPU in terms of the total communication cost reduction, as DPU has no impact on the edge-to-server communication involving newly collected data samples. This experimental setup assumes that all data samples in $\delta\mathcal{D}^r$ are collected by $N$ edge nodes during all rounds and sent to the server on a per-round basis. For clarity, let $S_d$ denote the data size of each training sample. During round $r$, we define per-node communication cost under DPU as $S_d \cdot |\delta\mathcal{D}^r|/N + (S_w \cdot k \cdot I + S_x(k) \cdot I)$. Due to space constraints, the detailed results are shown in Appendix D.3.1. We observe that DPU can still achieve a significant reduction on the total communication cost, *e.g.*, reducing up to $88.2\%$ on updating MLP and VGGNet even for the worst case (*i.e.*, a single node). Moreover, DPU tends to be more beneficial when the size of data transmitted by each node to the server becomes smaller. This is intuitive because in this case the server-to-edge communication cost (thus the reduction due to DPU) dominants in the entire communication cost.

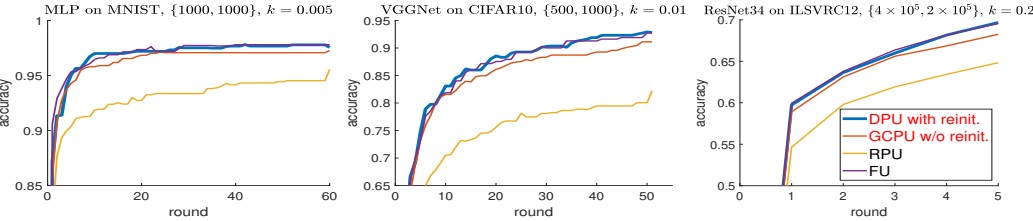

Figure 2: DPU is compared with other baselines on different benchmarks in terms of the test accuracy.

Table 2: The average accuracy difference over all rounds, the ratio of communication cost over all rounds related to full updating, and the number of rounds that are required to send partial updating.

| Method | Average accuracy difference | | | Ratio of communication cost (Updating rounds) | | |
|---|---|---|---|---|---|---|
| | MLP | VGGNet | ResNet34 | MLP | VGGNet | ResNet34 |
| DPU | $\mathbf{-0.17\%}$ | $\mathbf{+0.33\%}$ | $\mathbf{-0.12\%}$ | 0.0071 (22) | 0.0183 (35) | 0.2226 (5) |
| GCPU | $-0.72\%$ | $-1.51\%$ | $-1.01\%$ | 0.0058 (18) | 0.0198 (38) | 0.2226 (5) |
| RPU | $-4.04\%$ | $-11.35\%$ | $-4.64\%$ | 0.0096 (30) | 0.0167 (32) | 0.2226 (5) |

### 4.3    IMPACT DUE TO VARYING NUMBER OF DATA SAMPLES AND UPDATING RATIOS

**Settings.** In this set of experiments, we demonstrate that DPU outperforms other baselines under varying number of training samples and updating ratios. We also conduct an ablation study concerning the re-initialization of weights discussed in Section 3.2. We implement DPU with and without re-initialization, GCPU with and without re-initialization and RPU (see Section 4.2) on VGGNet using CIFAR10 dataset. We compare these methods with different amounts of samples $\{|\mathcal{D}^1|, |\delta\mathcal{D}^r|\}$ and different updating ratios $k$. Each experiment runs three times using random data samples.

**Results.** We compare the difference between the accuracy under each partial updating method and that under full updating. The mean accuracy difference (over three runs) is plotted in Figure 3. A comprehensive set of results including the standard deviations of the accuracy difference is provided in Appendix D.4. Note that the green curves in Figure 3 represent pruning methods that will be discussed in the next section. As seen in Figure 3, DPU (with re-initialization) always achieves the highest accuracy. The dashed curves and the solid curves with the same color can be viewed as the ablation study of our re-initialization scheme. Particularly given a large number of rounds, it is critical to re-initialize the start point $\boldsymbol{w}^{r-1}$ after performing several rounds (as discussed in Section 3.2).

In the first few rounds, partial updating methods (including random partial updating) almost always yield a higher test accuracy than full updating, *i.e.*, the curves are above zero. This is due to the fact

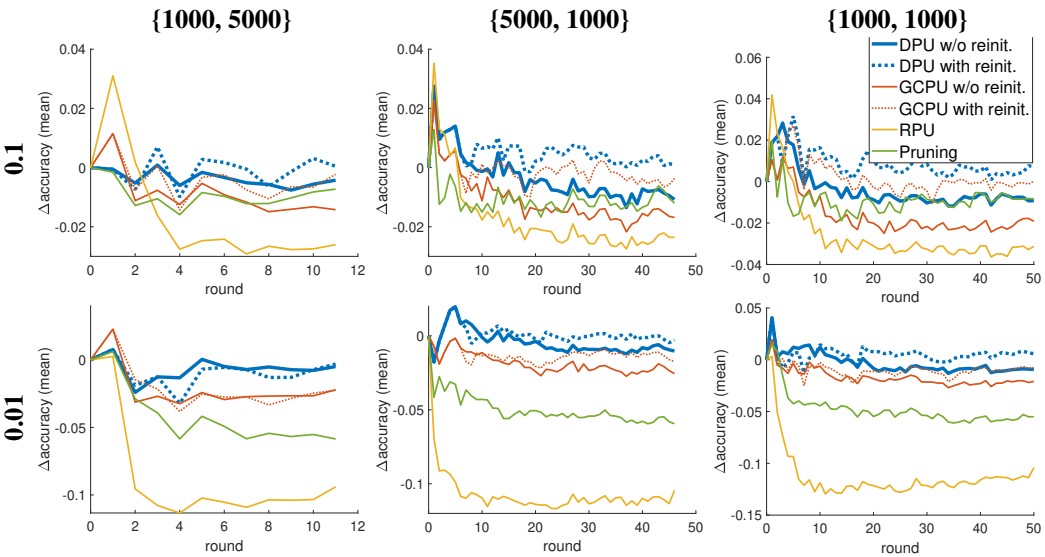

Figure 3: Comparison w.r.t. the mean accuracy difference (full updating as the reference) under different $\{|\mathcal{D}^1|, |\delta\mathcal{D}^r|\}$ (representing the available data samples along rounds, see in Section 4) and updating ratio ($k = 0.1, 0.01$) settings.

that the amount of available samples is relatively small in the first few rounds and partial updating may avoid some co-adaptation of weights which happens in full updating. This results in a higher validation/test accuracy. Note that the three partial updating methods perform almost randomly in the first round compared to each other, because the limited sample size (*i.e.*, $|\mathcal{D}^1|$) is not sufficient to distinguish between critical weights. This fact also motivates us to (partial) updating the first deployed model when new data are available.

## 4.4 COMPARING PARTIAL UPDATING WITH PRUNING

**Settings.** We compare the partial updating methods mentioned in Section 4.3 with a state-of-the-art pruning method proposed in (Renda et al., 2020), where the network is first trained from a random initialization at each round, then conducts one-shot magnitude pruning (set weights as zero), and finally, is sparsely fine-tuned with learning rate rewinding. The ratio of non-zero's weights in the pruning method is set to the same as the updating ratio $k$ to ensure the same communication cost.

**Results.** We compare the difference between the accuracy under each method and that under full updating. The mean accuracy difference (over three runs) is plotted in Figure 3. As seen, DPU outperforms the pruning method in terms of accuracy by a large margin, especially under a small updating ratio. Note that we preferred a smaller updating ratio in our context because it explores the limits of the approach and it indicates that we can improve the deployed network more frequently with the same accumulated server-to-edge communication cost.

Note that one of our chosen baselines, global contribution partial updating (GCPU, Alg. 1), could be viewed as a counterpart of the pruning method, *i.e.*, pruning the incremental weights with the largest magnitudes. By comparing GCPU (with or without re-initialization) with "pruning", we conclude that retaining previous weights yields better performance than zero-outing the weights (pruning).

## 5 CONCLUSION

In this paper, we present the weight-wise deep partial updating paradigm, motivated by the fact that continuous full weight updating may be impossible in many edge intelligence scenarios. We present DPU, which is established through analytically upper-bounding the loss difference between partial updating and full updating, and only updating the weights which make the largest contributions to the upper bound. Extensive experimental results demonstrate the efficacy of DPU which achieves a high inference accuracy while updating a rather small number of weights.

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
