# OpenReview forum: "Deep Partial Updating"
_ICLR.cc/2021/Conference — Reject_

### Official Review · AnonReviewer3 · 2020-10-28
**Looks a practical new approach but the justification may need improvement**

**Rating:** 6
**Confidence:** 3

**Review:**

This paper proposes a deep partial updating paradigm that can reduces the computational and communication cost on the edge devices by only updating most important weights in each round instead of a full update.  It also proposes metrics to select weights by global and local contributions and the experiment results show the efficacy of the proposed methods.

In summary, the method proposed in this paper looks practical and easy to implement, but the theoretical justification needs further clarification. I'm not sure about the significance of this paper as I'm not an expert in this area, so I prefer to leave this to other reviewers to decide.

In general, the paper is well written and easy to follow, and the motivation is sound.  However, the justification of the global and local contributions need to be clarified further. The inequality of Eq.(3) can hold only if f is L-smooth and convex,  which indicates the loss function is assumed L-smooth and convex. So what's the justification of the definition of global and local contributions  when the loss is non-convex which is the most common case in the experiments?  Without the theoretical justification, the global contribution that selects weights with largest values is basically as the same as pruning,  and the local contribution basically measures the changed loss caused by the update of a weight.  Although they may be still practical but the novelty is limited.

The experiment results show that the proposed method can obtain similar performance with the full updating but costs much less communication overhead.  It seems a very practical method in this area and the paper provides an interesting empirical study. The simple combination of global and local contributions outperforms each individual contribution, I'm wondering if authors have tried more other ways to combine them? And why this way is better?

One minor comment regarding the structure of the paper: as the initialization strategy plays an role in this method, it would be better to put the experimental results of comparing different initializations to the formal content, and the appendix can be put after the bibliography in one file.

################ Feedback to the authors' response ###############

As the authors have addressed some of my main concerns and provided nice extra experimental results, I will raise my score to 6.

---

> ### Author Response · Authors · 2020-11-10
> **Reply to Reviewer3 (Looks a practical new approach but the justification may need improvement)**
>
> We appreciate your review and your constructive feedback. Here are our replies and changes:
>
> * Eq.(3): Please note that the inequality of Eq.(3) holds true **as long as**$f$ is L-smooth, and L-smooth is essential to ensuring convergence of many gradient-based algorithms. In addition, L-smoothness is independent of convexity, see also Page 24-27 in Sec. 1.2.2 of (Nesterov, 1998). Most importantly, convexity is not a precondition of Eq.(3).
>
> * Difference to Pruning Method: (i) The objective is fundamentally different. (ii) The chosen rewinding metrics are different from the current pruning metrics (Han et al., 2016; Frankle & Carbin, 2019; Renda et al., 2020). (iii) We propose a re-initialization strategy considering a large number of updating rounds. Please see more details in our "General Reply to All Reviewers".
>
> * Combining contributions: We indeed tried some other alternatives, for example, taking the minimum of both normalized metrics, $\mathrm{min}(\frac{1}{\boldsymbol{1}^\mathrm{T} \cdot \boldsymbol{c}^\mathrm{global}} \boldsymbol{c}^\mathrm{global}, \frac{1}{\boldsymbol{1}^\mathrm{T} \cdot \boldsymbol{c}^\mathrm{local}} \boldsymbol{c}^\mathrm{local})$.  However, the most straightforward combination (adding them together) yields a more stable performance in the experimental results. Intuitively, local contribution can better identify critical weights w.r.t. the loss during training, while global contribution may be more robust for a highly non-convex loss landscape. Thus, both metrics may be necessary when selecting weights to rewind.  We added the corresponding information to the revised version.
>
> * Editorial issues: Due to page limitation, we put the comparison results with different initializations in Appendix D.1, D.2. In the final version, we will merge the supplementary file and the main text.

---

### Official Review · AnonReviewer4 · 2020-10-28
**Interesting work, hampered by presentation**

**Rating:** 6
**Confidence:** 3

**Review:**

Summary:  This paper presents a method to reduce the bandwidth required to update DNN models on edge devices.   The key insight is that model updates typically incorporate new data (training samples), and that after doing so, a minority of weights capture the majority of change due to retraining.  The authors propose a method by which to identify this weight subset, and compare the relative size (and test accuracy) of that update to that of other solutions (such as sending the entire network or sending a random subset of the weights on each retraining round).  Experiments with a number of existing data sets and models illustrate that the approach reduces update size more than 77% while maintaining reasonable test accuracy.

=== pros ===

+ Paper is sufficiently motivated.  As edge devices use more (and larger) models and as their count increases, the relevance of partial update techniques to accomodate model decay will remain.

+ The proposed technique provides a "sister" technique to pruning, not identifying nodes with greatest weights to retain, but identifying weights with the greatest changes to retain.  The policy is informed by choosing weights that minimize the difference in loss between the fully retrained network and its partially updated version.

+ The paper is rounded out by practical items, such as encoding weight ids, a policy to determine when to retrain the network from scratch
(re-initialization), and avoiding sending updates when validation error does not significantly change.

+ The evaluation looks at a variety of angles, the ratio of initial training set to update size, different data sets and architectures, and compares to a random partial update strategy as well as a simplified version of their approach "GCPU".

=== cons ===

- The overall presentation is difficult to parse.
- The technique owes much to pruning methods and methodologies.   The technical approach (choosing weights, iterative rewinding) follows from recent work on pruning.  It would be great to have that discussion in related work, moving it out of Section 3.1 and Section 4.
- Ultimately, existing pruning techniques can reduce networks by 90%.  By Amdahl's law, this implies that these techniques reduce communication by 7-10%, not 70-99%.
- Equally important, does the technique work well on pruned networks?  Unimportant updates may not be as available in such networks.  On the other hand, if you do the comparison and all updates are important, then over the course of the lifetime of the installed NN, using DPU instead of pruning would be the winner.
- Experiments in key graphs aren't clear: is there re-initialization in Figure 2?  Figure 3 performance never falls relative to full updating during re-initialization.  While the text (S3.2) makes it seem that the nodes reset all weights, using only 1% of the weights would impact test accuracy relative to full updating.

=== suggestions / questions ===

Overall, I found the work interesting, useful, and complete (aside from eval sec questions above).

It would be useful to introduce a metric that combines update size with accuracy loss at the beginning of the paper.  The evaluation does this, but consider pulling it forward and defining it explicitly.  Each round incurs a communication cost in bytes and experiences some accuracy, so, for example, one can capture changes in accuracy per byte, i.e. model improvement by update size.  Since you are comparing to other techniques that can reduce the bandwidth similarly, we want to optimize this ratio.

Some networks work very well with small k.   But how low can you go?   I.e., how does one choose k?   Perhaps the accuracy/bytes metric could be informative.

It would be interesting to discuss on why winning lottery ticket theory gets us 80-90% reductions, but this technique admits 99% reductions by retaining information in the rest of the network.

The startup procedure is not clear.  The graphs and discussion in S3.2 make it sound like the entire network re-initializes.  Can we be clear about what the first network looks like?  I'd assume all the weights.  But if we start from random values (sending the seed), the first round only updates $kI$ weights.  Can the test performance of the network with only 1% of its weights be 65% (Figure 2)?  Similarly, if DPU is re-initializing, why is the test accuracy monotically increasing -- the installed network would go back to ground zero.  Clearly I'm missing something, or your measuring the performance of the network at the server (w^f) and not the installed network (w^r).

Similarly Table 2 should have a column for the number of rounds that required no communication.

DPU won't send updates if validation error doesn't decrease significantly.  It isn't clear whether you gave the same benefit to Full Updating.

=== writing / terminology / notation ===

Overall the presentation is difficult to get through.   For instance, Section 3 has many awkward constructions.  It seems like there's a simple picture here, similar to the Train, Prune, Retrain flow of pruning work.  It seems deeply analogous, with the exception that rewinding replaces pruning.  The evaluation section refers to Alg 1 and Alg 2, but Section 3.0 refers only to "step 1" and "step 2".   Are there better words than step?  This section also refers to the second step as an "optimization" step.   You end P1 by saying you're optimizing eq 1 in step 2, then you say step 1 optimizes the same equation.  The last sentence of the S3.0P3 re-iterates what was said in P1.  I'm sorry, but it's a bit of a slog.

The use of notation is consistent.  There are a couple of things that felt like speed bumps.   I kept wanting to parse \delta W and \delta D using \delta as a variable, like $kI$.  At the end of section 2, introducing a new form of w^r as w~, was confusing.  Do we need w~?

Sometimes you use L0 norms (S2 eq 2) and other times you use summation (S3.1).

You use curly braces S4P1 for the sizes of the initial training and updates.  It looks like a set, not a configuration.  The text says the two sizes "represent the available data samples."  But here it's just a configuration -- it's not the set of samples at all (and it wouldn't be b/c not all updates R are present).

==== nits ====

Please learn the difference between that and which.   Remove "in order to."   Capitalize Figure and Section.

Some references use name style, other use indices. But the bibliography is all by name.  #confused.

---

> ### Author Response · Authors · 2020-11-10
> **Reply to Reviewer4 (Interesting work, hampered by presentation)**
>
> Reviewer comments “cons”:
>
> * Discussing Relation to Pruning and Rearranging Paper: We agree that our work is inspired by recent pruning papers, e.g., (Renda et al., 2020). Thus, we have included a discussion in the related work and a comparison w.r.t. SOTA pruning methods in Sec. 4.4 in the revised version, also see the comment "General Reply to All Reviewers".
>
> * Amdahl's law: In case of pruning and partial updating, only the non-pruned weights or updated weights need to be communicated. In analogy to Amdahl, there is of course some part that cannot be reduced: communicating which weights are updated, i.e., their indices. This is discussed in detail in Sec. 4.0 in the paper and is taken into account as overhead.
>
> * Does the Technique Work on Pruned Networks: There are two possibilities when conducting DPU on pruned networks, (i) the zero weights are fixed, (ii) the zero weights are allowed to be updated.
>
>  For (i), DPU may not work well if the original pruning ratio is large, since those important weights (non-zero weights) are selected based on a small number of samples. With more collected samples, the weights' importance will change.
>
>  For (ii), DPU is orthogonal to pruning. The original pruned network acts as a different startup network in comparison to our used random initialization. But note that if the original pruning ratio is quite large, the zero's weights need to be re-initialized to break the symmetry.
>
> * Unclear Re-Initialization: The "DPU" in Fig. 2 stands for Alg. 2 with re-initialization. We have adjusted the legend in the revised version for clarification. The accuracy drop caused by re-initialization could be better observed in Fig. 5 given in Appendix D.2. Each time the network is re-initialized, the new partially updated network may suffer from an accuracy drop. However, we can simply avoid such an accuracy drop by not updating the network if the validation accuracy does not increase compared to the last round. After implementing the above strategy, we plot the mean accuracy in Fig. 7. We have moved this discussion from Appendix D.2 to Sec. 3.2. This kind of accuracy drop could also be observed in Fig. 3 if simply comparing the branch point between the solid curve (without re-initialization) and the dashed curve (with re-initialization) with the same color.
>
> Reviewer comments “suggestions/questions”:
>
> * Metric of $\Delta$accuracy/bytes: We think this could be used as a new evaluation angle, yet requiring further consideration. During certain rounds, where the validation error of the model does not decrease, the server does not send any updates to the edge. Therefore, $\Delta$accuracy/bytes (i.e., 0/0) does not have a mathematical definition in such rounds, whereas the metrics we reported in Table 2 avoid this problem, i.e., (i) the accuracy difference related to full updating averaged over all rounds, and (ii) the ratio of the communication cost over all rounds related to full updating.
>
> * Updating Ratio: See our "General Reply to All Reviewers".
>
> * Retaining Weights vs. Zero-Outing Weights: See our "General Reply to All Reviewers" and Sec. 4.4 in the revised version.
>
> * Startup Network: The startup network, namely $\boldsymbol{w}^0$, is **entirely**randomly initialized. We report the performance of the installed network $\boldsymbol{w}^r$ in Fig. 2. We added more explanations in the revised paper. The network of the first round indeed only updates $k \cdot I$ weights from random initialization. But the observed test accuracy of 65% is achieved at the third round, i.e., the network could yield such an accuracy level with (maximum) 3% of updated weights from random initialization given 2500 samples. This is because we cropped the accuracy range in Fig. 2 to clearly demonstrate the difference between multiple curves. For the first round's performance, we refer the reviewer to look at the $\Delta$accuracy plots or the subplots of {5000,1000} in Fig. 13. We added more discussion about the performance of the first few rounds in Sec. 4.3.
>
> * We have included extra columns for the number of rounds that need server-to-edge communication in Table 2.
>
> * For a fair comparison, all compared methods (including full updating) will not send updates if the validation error does not decrease related to the last round. We have further explained it in the revised version. The horizontal straight line segments in Fig. 2 represent those non-updated rounds under each method.
>
> Reviewer comments “writing / terminology / notation / nits”:
>
> * Thanks for the comments. We have included all your suggestions as much as possible in the updated paper, in particular, (i) making a more clear clarification of Sec. 3.0 in the revised version; (ii) changing all $kI$ to $k\cdot I$; (iii) using L0-norm for consistency.
>
> * We fixed grammar and formatting issues.

---

### Official Review · AnonReviewer2 · 2020-10-28
**A new application scenario with minor technical contribution**

**Rating:** 5
**Confidence:** 3

**Review:**

Summary：
The paper proposes a weight-wise partial updating paradigm which adaptively selects a subset of weights to update at each training iteration while achieving comparable performance to full training. Experimental results demonstrate the effectiveness of the proposed partial updating method.

Strengths:
1.  The paper is well written.
2.  The process of upper-bounding the loss difference is clear.
3.  Experiments are conducted on various datasets with various net structures to support the proposed method.

Weakness:
My major concern is about novelty and contribution. Although the paper show some application scenarios of partial updating, I still think that pruning would be more proper. Furthermore, the metric of global contribution and local contribution is quite like choosing two similar weight norms to select top-k weight, which is very similar to pruning tasks. So I suggest rejecting this paper.

----
The authors’ rebuttal and the revised version have not fully addressed my concerns. It is not surprise that partial updating outperforms pruning by a large margin, as the inference of small updating still uses the whole weights of the network. Comparing to pruning, the technical contribution of this work is limited, so I would like to keep my original rating.

---

> ### Author Response · Authors · 2020-11-10
> **Reply to Reviewer2 (A new application scenario with minor technical contribution)**
>
> We appreciate your review and your constructive feedback. Here are our replies and changes:
>
> * Novelty: To the best of our knowledge, this is the first work on studying iteratively partial updating deployed networks using newly collected data. We introduce a favorable method for the proposed question. Partial updating targets reducing the server-to-edge communication cost when updating the deployed networks in multiple updating rounds, which is a fundamentally different objective from pruning. Also, partial updating is more proper and better in this scenario. Instead of zero-outing some weights, we can leverage some learned knowledge by retaining some previous weights. See our "General Reply to All Reviewers".
>
> * Pruning vs. Deep Partial Updating: We conduct some comparison experiments w.r.t. the SOTA pruning method (Renda et al., 2020), see more details in our "General Reply to All Reviewers" and Sec. 4.4 in the revised version. In conclusion, our DPU outperforms the pruning method in terms of accuracy by a large margin, particularly under a small updating ratio.
>
> * Differences in Methods: (i) The chosen rewinding metrics are different from the current pruning metrics (Han et al., 2016; Frankle & Carbin, 2019; Renda et al., 2020). The magnitude pruning uses the L1-norm of weights to select top-k weights, while the metric of global contribution (Alg. 1) uses the L1-norm of incremental weights to select top-k weights. The metric of local contribution uses the gradient-based information gathered during the first step of Alg. 2. This metric does **not**have an explicit relation to weight norms. This is also explained in the revised version. (ii) We propose a re-initialization strategy considering a large number of updating rounds.

---

### Official Review · AnonReviewer1 · 2020-10-29
**Easy to follow, and want to see more analysis and details**

**Rating:** 6
**Confidence:** 3

**Review:**

In this paper, the authors have proposed a new approach to determine the optimized subset of weights instead of simply conduct full weights updating. In order to better update the weights, they measure each weight's contribution to the analytical upper bound on the loss reduction from two sides (global and locally). After evaluation, a weight will be updated only if it has a large contribution to the loss reduction given the newly collected data samples. The experimental results show that their method can achieve a high inference accuracy while updating a rather small number of weights.

Strength:
The idea is easy to follow and seems applicable to be adopted.
Paper is well structured and written in general.

Weakness:
1. Lack of explanations:
	(1) from reward measurement side (motivation side):
In the introduction, the authors did not explain why they pick the loss as the weight measurement criteria instead of others (e.g., accuracy). While they report the accuracy in the evaluation part as one evaluation results.
	(2) from the update algorithm side:
The paper did mention their weights updating method is determined via both global and local contributions, and they talked in 3.1 'It turns out experimentally, that a simple sum of both contributions leads to sufficiently good and robust final results'. however, it is not convincing that those two facts can have the equal impacts on the final prediction.
	(3)  from the updating setting side:
It seems that the defined updating ratio is one important factor as discussed in section2, not  enough contents are provided in the paper to describe how to calculate this ratio.
	(4) re-initialize mechanism:
Re-initialize is also another important factor in the weight updating as discussed in section 3.2 'trained from the last round for a long sequence of rounds. Thus, we propose to re-initialize the weights after a certain number of rounds', however, the computation of how many rounds the network needs to be re-initialized seems not plausible.
2. Evaluation:
	(1) lack of comparison: It would be good if authors can apply their method on some recent works (or models), which can also show others how flexible their method can be adopted or applied
	(2) there is no contents in the paper showing how authors decide their experiment settings, for example, why authors always select k (weight changing ratio) as very small 0.01, 0.05, 0.1, 0.2 instead of 0.5
	(3) in Fig2, it is curious why authors apply different settings on different datasets when comparing their methods
	(4) for section 4.2, it would be good if the authors can also try other initialization ways, for example using the average weights in each round window instead of directly using the latest round weights
	(5) in Table 1, it seems full updating still can beat the combined method, however, in Fig2, authors did not explain why DPU has better performance than other settings even compare with the full update
	(6) in Fig3, while DPU with re-init can achieve best performance than others, there is no explain about why it did not perform well in the first few rounds
	(7) the authors did not mentioned how many runs which they have conduct their experiments to provide the results
3. Some parts need to be further improved for example
	(1) Fig3, it would be good if authors can add some texts information for {1000, 5000};
	(2) Section3 is a little bit hard to follow need to be reorganized
	(3) Related work can be further improved to better cover most recent works

---

> ### Author Response · Authors · 2020-11-10
> **Reply to Reviewer1 (Easy to follow, and want to see more analysis and details)**
>
> Thanks for the reviewer's constructive feedback. However, we appreciate it if the reviewer could provide details
> and references in some comments.
>
> * 1.(1) Like most learning methods, the algorithmic part of the paper is based on minimizing a loss function. Accuracy is only used for comparison of classification tasks, but our method also targets some other learning tasks, e.g., language modeling (see Appendix D.3.2), where accuracy is inapplicable. An explanation has been added to the revision.
>
> * 1.(2) We agree that those two facts may have different impacts on the final prediction. We thus normalize each contribution value according to its significance in its own set (either local contribution set or global contribution set), see Eq.(10). The corresponding discussion is added to the revised version.
>
> * 1.(3) See our "General Reply to All Reviewers".
>
> * 1.(4) The mechanism of after how many rounds the network needs to be re-initialized is based on experimental results and the corresponding reasoning, given in Appendix D.2.
>
> * 2.(1) To the best of our knowledge, this is the first work on studying weight-wise partial updating a network
> using newly collected data in iterative rounds. Therefore, we developed several baselines for comparison
> (also see in Sec. 4.2), including extending a SOTA unstructured pruning method (Renda et al., 2020). We would appreciate it if the reviewer could provide any related work that needs to be added to the comparison.
>
> * 2.(2) See our "General Reply to All Reviewers".
>
> * 2.(3) We note that the settings of different benchmarks cannot be compared, because different benchmarks have different amounts of samples in datasets; different datasets also incur different dimensionalities of each sample, that require different neural network architectures and different optimizers during training. Therefore, we use the same setting for comparing different methods on a single benchmark but may change for different benchmarks. More experimental results regarding different updating ratios and different numbers of data samples are provided in Sec. 4.3, Appendix D.4.
>
> * 2.(4) (i) Sending the average weights using the communication link is not useful due to the communication constraint; (ii) doing an averaging on the device needs more storage as previous (aggregated) weights also need to be remembered locally.
> The conducted (re-)initialization methods, random (via sending a random seed) and last round (without communication cost), are the two most communication-efficient methods. Therefore, other approaches have not been tested experimentally. This discussion has been added to the paper.
>
> * 2.(5) Table 1 shows the ablation results of different rewinding metrics, in terms of the incremental training loss caused by rewinding. Thus, these loss values are only intermediate results during partial updating. We added more explanation to the text in order to avoid such a misunderstanding.
>
> * 2.(6) In Fig. 3, three partial updating methods perform randomly in the first round compared to each other due to the limited sample size, which is not sufficient to distinguish between critical weights. The variance of the first few rounds is also quite large, see Fig. 12. We explained this more thoroughly in the revised version.
>
> * 2.(7) Each experiment runs three times using random data samples, as stated in Settings of Sec. 4.3. More experimental details, as well as the standard deviation plots, can be found in Appendix D.
>
> * 3 We followed the suggestions of the reviewers and added more information. We would appreciate it if the reviewer could provide some references as we already did a very extensive literature review about relevant recent work.

---

### Author Response · Authors · 2020-11-11
**General Reply to All Reviewers: Deep Partial Updating vs. Pruning & Updating Ratio $k$**

Dear reviewers,

We appreciate all reviewers' constructive comments. We have uploaded a revised version of our paper containing results on pruning vs. deep partial updating. The changes are marked in red color.

We have noticed that the following concerns are raised in multiple reviews: (i) whether our proposed partial updating may significantly outperform the state-of-the-art pruning methods; (ii) how to choose the updating ratio $k$.

Concerning (i), we conducted additional comparison experiments and provide more information:

*  Most importantly, additional experiments using a state-of-the-art magnitude pruning method proposed in (Renda et al., ICLR 2020)  are reported in the updated pdf (Sec. 4.4). The results show that our DPU outperforms the pruning method in terms of accuracy by a large margin, particularly under a small updating ratio.

* The objectives of traditional pruning and partial updating are fundamentally different. Traditional pruning methods (Han et al., 2016; Frankle & Carbin, 2019; Renda et al., 2020) aim at reducing the number of operations and storage consumption. Our partial updating paradigm targets reducing the server-to-edge communication cost when updating the deployed networks.

* By retaining some previous weights while updating the others, we can leverage some learned knowledge for better performance with the same communication cost.

* Pruning requires more on-device memory access (also more energy consumption as discussed in Sec. 1) than partial updating.

Concerning (ii):

* The updating ratio is determined by the communication constraints in practical scenarios, as we also explain in Sec.1. For example, if the communication bandwidth only allows the transmitted data size from the server to each edge device to be smaller than 10MB in each round, assuming a 100MB data size of all weights, the updating ratio could be set to 8%~9% considering the extra indexing cost.

* Note that we preferred a smaller updating ratio in our context because it explores the limits of the approach and it indicates that we can improve the deployed network more frequently with the same accumulated server-to-edge communication cost.

---

> ### Author Response · Authors · 2020-11-22
> **New Version Uploaded**
>
> Dear reviewers,
>
> We have added more comparison results w.r.t. pruning (Renda et al., ICLR 2020) in Fig. 11~Fig. 14 in Appendix D.4.

---

### Decision · Program_Chairs · 2021-01-07
**Final Decision**

**Decision:**

Reject

**Comment:**

The paper proposes an approach to selectively update the weights of neural networks in federated learning. This is an interesting and important problem. As several reviewers pointed out, this is highly related to pruning although with a different objective.  It is an interesting paper but is a marginal case in the end due to the weakness on presentation and evaluation.